# Negotiating Complexity: Challenges to Implementing Community-Led Nature-Based Solutions in England Pre- and Post-COVID-19

**DOI:** 10.3390/ijerph192214906

**Published:** 2022-11-12

**Authors:** Yichao He, Anna Jorgensen, Qian Sun, Amy Corcoran, Maria Jesus Alfaro-Simmonds

**Affiliations:** 1Department of Landscape Architecture, University of Sheffield, The Arts Tower, Sheffield S10 2TN, UK; 2School of Design, Royal College of Art, Kensington Gore, London SW7 2EU, UK; 3Just Futures Centre for Child, Youth, Family and Community Research, School of Human and Health Sciences, University of Huddersfield, Queensgate, Huddersfield HD1 3DH, UK

**Keywords:** nature-based solutions, green social prescribing, health and wellbeing, COVID-19

## Abstract

Nature-based solutions (NbS), including green social prescribing (GSP), are sustainable ways to address health and wellbeing, especially since the COVID-19 pandemic exacerbated the strain on healthcare. NbS require national and local cross-sector coordination across complex, interrelated systems, but little is known about the specific challenges this poses for community-led NbS. We carried out a traditional literature review to establish the context and knowledge base for this study and interviewed 26 stakeholders. These came from environment, health and social care sectors at national and local levels, with local-level stakeholders from Bradford and Walsall: English cities significantly affected by the pandemic, with high levels of deprivation and health inequality. The interviews explored experiences of implementing NbS, both pre- and post-pandemic and the resulting renewed interest in the salutogenic effects of engaging with natural environments. We coded the interview transcriptions using NVivo to identify the challenges existing in the systems within which these stakeholders operate to create and manage NbS. By synthesizing what is known about the challenges from existing literature with findings from the interviews, we developed eight categories of challenges (perception and knowledge, political, financial, access to natural spaces, engagement, institutional and organisational, coordination, GSP referral and services) faced by multiple sectors in implementing community-led NbS in England. Furthermore, this study highlights the new challenges related to the pandemic. Identifying these challenges helps stakeholders in existing complex systems recognise what is needed to support and mainstream NbS in England.

## 1. Introduction

Personalised care is a new service model for the National Health Service in England (NHS), meaning people have control over how their care is planned and delivered, based on individual needs and preferences [1]. Social prescribing, as a component of personalized care, generally involves referring individuals or households from health care settings to non-clinical community services provided by the voluntary community and social enterprise sectors (VCSE), to participate in activities to improve their physical, mental and social health and wellbeing [2]. It was first popularized in the UK and has been implemented in more than a dozen countries over the last decade, including Australia, Brazil, Canada, China, Denmark, Finland, Japan, Ireland, the Netherlands, the UK, USA, New Zealand, Portugal, Singapore, South Korea, Sweden, Spain [3,4,5,6,7]. As a type of social prescribing, green social prescribing (GSP)—defined as “the use of nature-based activities for improving physical and mental health” [8]—is supported by government policy in England [9], with a GBP 4 million investment in 2020 to embed GSP into communities [10]. It is recognised as an effective intervention for a range of health conditions and wellbeing issues, especially relating to mental health [11,12,13]. Nature-based solutions (NbS)—more generally defined as actions inspired by, supported by, or copied from nature, that aim to address environmental, social and economic challenges [14]—also help maintain wellbeing in the general population, as well as manage mental disorders [15,16]. NbS is an umbrella concept involving the creation of new green spaces and green space engagement programmes aimed at the general population, as well as a range of GSP offers, including activities such as park visits, walking clubs, gardening, forest bathing, and bird watching [17]; all based on the health and wellbeing benefits of nature. Mental health issues have increased greatly during the COVID-19 pandemic [18] and it has therefore never been more critical to maximise GSP and NbS capacity. 

In England, the VCSE sector is the foundation of GSP and local NbS, in terms of actually providing GSP, and animating and championing local green spaces. In this study, community-led NbS refers to nature-based actions that are organised and managed by VCSEs in close collaboration with local communities, the latter being integral to connecting people effectively with nature for health and wellbeing benefits pre- and post-pandemic. If access to nature is to be readily available to those with a defined need, as well as the general population, then it is vital to understand this sector’s needs. This paper, therefore, focuses on exploring the challenges to implementing NbS (including GSP) from the VCSE perspective in a world dealing with the realities of COVID-19.

### 1.1. Public Health Emergency

A staggering one in four adults in England experience at least one diagnosable mental health issue per year [19,20], and one in six are diagnosed with symptoms of common mental disorders, e.g., generalised anxiety disorder, and depression [21]. The COVID-19 pandemic has seen an escalation in mental health issues. Significantly higher percentages of depressive, anxiety and insomnia symptoms were identified in the UK during the lockdown in March 2020, compared with pre-pandemic levels [18], and the percentage of people experiencing anxiety during the pandemic almost doubled from pre-pandemic levels in the UK [22].

The high prevalence of common mental disorders has a significant cumulative cost to society [22]. In 2020/21, mental health services cost the National Health Service (NHS) in England GBP 14.3 billion: 14.8% of local NHS funding allocations [23]. A more recent report states that mental health issues conservatively cost the UK economy GBP 117.9 billion per year, equating to 5% of the UK’s GDP in 2019 [24]. Mental health issues also cause loss of productivity from reduced workforce engagement [25], and lost income taxation revenue [26,27].

The pandemic exposed fundamental weaknesses in the NHS in England, as well as the neglect of social care by successive governments, persistent and wide health inequalities accentuated by cuts to public health budgets, and growing pressure on services due to the scaling back of care for non-COVID conditions [28]. There is therefore an urgent need to address mental health care during and beyond the COVID-19 pandemic and to build a system of care to address the weaknesses in the current NHS system that COVID-19 exposed.

### 1.2. Benefits of Natural Environments and NbS

An array of evidence shows that access to and use of green space supports mental health benefits, including restoration from stress [29], reduced psychological distress [30] and decreased depression and anxiety [31]. These mental health benefits are also identified in more targeted horticultural therapy [32]. Green space benefits are often intermingled; for example, community gardening involves physical activity, supports restoration from stress, and promotes social contacts [33], whilst visiting green spaces promotes physical activities and mitigates feelings of loneliness, indirectly contributing to mental health improvements [34]. The review of 82 studies focusing on NbS for vulnerable youths reported largely positive psychosocial and behavioural outcomes, often maintained post-treatment [35]. Furthermore, a series of case studies from the Ecominds programme of nature-based health interventions for mental health and wellbeing was estimated to have saved an average of GBP 7082 per participant per year, through reduced NHS costs, benefits reductions as a consequence of moving into employment, and increased tax contributions [36].

The number of people visiting parks in the UK and worldwide increased during the COVID-19 pandemic [37,38,39]. Urban green spaces were the most popular natural spaces in England during this time [40]. Green space had benefits for mental health during the COVID-19 pandemic [41,42,43,44]. Access to pleasant outdoor space was a determining factor for feeling less stressed and tired and having a less disrupted lifestyle, fewer health concerns, and a more positive outlook [45]. Urban green space became a significant place for people to find solace, respite, exercise, and relaxation during periods of COVID-19 isolation [46]. The positive psychological, physical, social, and spiritual effects of green space made it one of the only sources of resilience [37]. Proactive interventions to increase participation in nature-based activities could help build resilience prior to global crises [47]. The aforementioned benefits highlight the potential for implementing NbS to support mental health and public health more widely in future.

Even though nature surged in popularity during COVID-19, providing people with vital health and wellbeing benefits, studies report that deprived areas have poorer accessibility to green space [48,49,50], or worse quality of green space, compared with more privileged groups [51,52]. Inequalities in visiting green spaces were sustained during the lockdown period and are now entrenched [38,53,54]. The factors such as psychosocial stress and subsequent risk-taking behaviours, reduced access to health-promoting facilities and services, occupational health risks, psychosocial factors, and health-related behaviours, also explained the reason for lower socio-economic groups’ health inequality [55]. The reduced access to healthcare during the COVID-19 worsened outcomes in disadvantaged areas and marginalised communities [56]. In the US, block groups with lower income and the majority people of colour are less green and have fewer parks, and communities with the least nature nearby are the ones most impacted by COVID-19 [57]. There is, therefore, an urgent need to find ways of addressing these inequalities.

### 1.3. Government Interest in NbS

The need for high-quality, accessible urban green spaces, as well as programmes to encourage people to use them, is set out in national policy in England. The National Planning Policy Framework [58] promotes the creation of green infrastructure for a wide range of benefits, including health and wellbeing. The 25-Year Environment Plan [9] pledges to connect people “systematically” with green spaces for preventative and therapeutic purposes, with a particular focus on improving mental health, as well as expanding green infrastructure and setting new quality standards, focusing especially on areas where there is a deficit in quantity or quality.

In 2020, the British government invested GBP 4 million in piloting approaches to green social prescribing (GSP) in order to “improve mental health outcomes, reduce health inequalities, reduce demand on the health and social care system, and develop best practice in making green social activities more resilient and accessible” [59]. VCSEs are seen as central to the provision and scaling up of programmes to connect people with nature, including GSP. However, the NHS guide to social prescribing and community-based support emphasises the extensive resource needed by the VCSE sector if it is to play a significant role in GSP, including both funding and the need for close collaboration with other relevant organisations and infrastructure, including local authorities and health and social care providers [60]. Some commentators question whether the current push for GSP takes sufficient account of the needs of local communities and the challenges faced by VCSEs, not least the need for funding and long-term sustainability, and emphasise the importance of an approach centred on understanding the sector’s and users’ needs [61].

This paper is based on the Nature’s Way project—a research study carried out by the Royal College of Art and the University of Sheffield. The initial phase of the Nature’s Way project explored the challenges and enablers in implementing NbS through a literature review and stakeholder interviews, focusing especially on community-led projects. The subsequent design phase involved collaborative work with local VCSE groups in Bradford and Walsall (a city and town in England, respectively), to co-create NbS knowledge. The final phase of the project developed a platform to disseminate and outreach the co-created NbS knowledge to empower communities, organisations, and individuals to innovate NbS as alternatives for healthcare and societal resilience following the COVID-19 pandemic.

The following sections will introduce our research questions and methods (Section 2), set out the review of the literature and relevant findings from the Nature’s Way project (Section 3), before finally summarising the contribution and implications of this research (Section 4).

## 2. Research Questions and Methodology

### 2.1. Research Gaps and Questions

This study focuses specifically on the challenges facing community-led NbS for health and wellbeing in England, especially in the context of the COVID-19 pandemic. To further explore the challenges surrounding the VCSE sector in this complex scenario, the data collection and analysis were driven by the following research questions:What are the challenges in implementing community-led NbS for health and wellbeing in the context of England?How has COVID-19 impacted these challenges?What are the new emerging challenges compared with those identified in previous research?

### 2.2. Data Collection and Analysis

This paper reports on the challenges identified in the initial phase of the Nature’s Way project, including a traditional literature review to establish the context and knowledge base of the project, and stakeholder interviews.

We used a range of search terms to retrieve the literature, i.e., nature-based solutions, social prescribing, social innovation, social care, green care, wellbeing, service design, landscape architecture, co-design, place-based, human-centred, COVID, stakeholders interviews, green prescribing challenges, community gardens challenges. We also followed up suggestions or reference lists for additional texts where relevant. We then scanned our collection of literature for studies focused on community-led NbS, implementation of blue-green infrastructure, nature-based social prescribing or green social prescribing, and NbS aimed at addressing key societal challenges, as these are areas with significant overlaps with the concept of NbS used in this research. The challenges are categorised differently across the literature, so we synthesised these categories into a revised classification that is also relevant to the findings from the Nature’s Way project.

Twenty-six participants from different national and local (in Bradford and Walsall) sectors were approached for formal interviews in a process ethically approved by the Royal College of Art (Table 1). These participants were identified using purposive and snowball sampling approaches and included stakeholders from the health and environmental sectors, local authorities, non-government departmental public bodies, funding bodies, the VCSE sector, NHS staff, social housing, local communities, and professional bodies. The stakeholders were approached by emails in the first instance. The email included a brief description of the project and the aim of the interview. Whilst we had an overall positive response, some stakeholders (*n* = 8) declined their participation due to time constraints. However, in all cases, they referred us to an alternative contact within their organisations. All participants gave informed consent to be interviewed. The interviews were semi-structured, covering the main themes relevant to the research, as well as enabling participants to explore issues important to them [62]. The interviews were conducted by experienced research associates from July 2021 to February 2022, remotely via Zoom at a quiet office or home setting due to the pandemic and the fact that many stakeholders were dispersed around England, or face-to-face on-site at the participants’ NbS projects when possible. The main themes comprised interviewees’ roles and their relationship with mental health and wellbeing, and NbS; their perceptions of the challenges facing the local community in terms of public health and wellbeing in the context of COVID-19; their organisation’s experiences with NbS; and the impact of COVID-19; as well as more targeted questions according to their role (see the Appendix A for the interview questions). The interviews were audio-recorded and automatically transcribed using the Otter software package. Relevant sections of the interview transcriptions were checked for accuracy against the original recordings and corrected where necessary. We had no dropouts, and all the interviewees seemed keen to be updated on the results of the project.

Transcriptions were imported into the Nvivo 12 Pro software package and analysed following the qualitative content analysis (QCA) method, using both inductive and deductive approaches commonly used in QCA [63], with the coding categories generated from the data or relevant previous research [64]. The first step was reviewing the transcripts and coding according to the research questions in an inductive approach to generate initial themes and sub-themes. The second stage was comparing these themes with the challenges identified in the existing literature to deductively generate a new classification of the challenges relevant to the Nature’s Way project findings, whilst building on the typologies found in the literature.

## 3. Results and Discussion

This section details the thematic challenges structured under the synthesised classification from the review of literature and findings from the interviews (Table 2, see the Appendix A for the codebook and quotation examples). The challenges and impacts of COVID-19 identified by the interviewees are set out and discussed under each theme.

### 3.1. Perception and Knowledge

Notable challenges in existing literature relating to perception and knowledge are a lack of understanding and awareness of NbS and their benefits among both decision-makers and the wider public [65,66,67,68]. Examples include a lack of awareness of NbS’s diverse environmental, social, and economic benefits [69], or GPs not seeing the health benefits of nature-based social prescribing [70,71]. These uncertainties regarding NbS benefits could result from a lack of evidence. Decision-makers may be discouraged from supporting NbS due to a lack of clinical evidence about the efficacy of green prescriptions, a lack of an explanatory framework for green care [72] or even a lack of evidence about local plant species with bioremediation potential and air quality benefits [69]. Another impediment is a lack of experience and knowledge about the development, maintenance and monitoring of NbS [69,71], or green social prescribing (GSP) providers’ lack of skills in dealing with severe mental health issues [70]. The lack of consistent terminology to describe the field is also identified as a knowledge challenge when promoting nature-based interventions and programmes [72,73]. Another related obstacle is path dependency [74], defined as past experience with a controlling influence on current decision-making [75], e.g., reluctance to implement NbS projects due to lack of relevant experience [76], negative experiences or fear of failure [77,78].

The challenge most frequently highlighted by the interviewees under this theme was the lack of awareness of the efficacy and cost of NbS among the general public, decision-makers, funding providers, and social care and education sectors, and the resulting difficulties in securing a prioritisation of supportive policy and funding for NbS, as well as getting GPs to prioritise GSP. The existing literature highlights this perceptual barrier as a particular challenge to mainstreaming NbS [79], articulated here by one of our interviewees:


*I think that perhaps one of the barriers is not recognising the value of it [GSP] and in a financial way, in a monetary way.*
(National, Non-departmental Public Body, Senior Adviser)

Interviewees also mentioned a lack of awareness among the general public: potential users may not see the benefits of NbS, may misunderstand NbS as social services, or may not trust social prescribers’ recommendations for GSP. Therefore, the need for perceptual change applies to the general public, and healthcare professionals [80]. Additionally, inconsistent terminology could present further difficulties in promoting awareness of NbS and GSP, as interviewees claimed that stakeholders cannot reach agreement on how to name these activities.

Finally, lack of specific knowledge and relevant experience in implementing NbS and GSP among some NbS providers, e.g., in relation to mental health, horticulture, safe use of equipment, as well as skills in writing successful funding bids, are also challenges identified in this study. When local organisations face issues beyond their capacity and are unable to deal with them, the negative experiences may cause them to be wary of participating next time; referred to as path dependency in previous research [75,77,78]. 

### 3.2. Political

Minor policy changes could contribute to a deeper paradigm shift for a healthier relationship between humans and the natural environment [81]. Legislation and policy can be effective in supporting NbS [74]. However, the aforementioned lack of awareness of the benefits of NbS leads to a political barrier: a lack of sense of urgency in implementing NbS among policy-makers [77,82,83]. Together with competing priorities between policies [76,84], lack of political will and long-term commitment also become notable political barriers resulting in a lack of supportive legal and policy frameworks [67,69,78,83]. Gaps between government policy and practice are also highlighted, making it difficult for local actors to implement projects [68,69]. Additionally, there is a lack of national-level authoritative voices to provide guidance on nature-based social prescribing and to coordinate activity [70]. Combined, these political barriers constitute a fundamental challenge to implementing and mainstreaming NbS.

A chief executive of a regional NbS provider in this study stated that there is a lack of national-level organisations that advocate for the local sector and provide funding streams for local organisations implementing nature-based activities. There is also a political need and challenge to move the medical model from cure to prevention, to help bring the wellbeing impacts of nature into the mainstream: a number of interviewees argued that current medical and social care models are not sustainable:


*I think that we have to look at the way we invest in traditional healthcare, to make the investment in prevention in the first instance cheaper than investing in the cure … So, one of the biggest hurdles is getting the decision-makers to recognise that importance.*
(National, Built Environment Professional Body, President)

### 3.3. Financial

Finance is the constraint most commonly mentioned in the surveyed literature [67,71,76]. Lack of sustained or dedicated financial resources was reported as the main financial challenge in maintaining the viability of projects or activities, and for mainstreaming [69,70,78,85,86], with resulting project insecurity also potentially impacting participants’ mental health [70]. The funding system for green social prescribing (GSP) is also highly complex, with multiple pathways and sources of funding. Kimberlee et al. (2022), for example, identify six different social prescribing funding models involving health care funding, not including other models involving funding from local authorities or the charitable sector [87]. The coordination of public and private finance is also a related barrier [88]. Whilst NHS and charitable funding may include payment to the provider, Dayson and Batty (2020) found that such payments seldom cover the full cost of GSP provision borne by VCSEs, which must, therefore, be cross-subsidised by other income streams [89]. It is onerous and time-consuming for small GSP providers to negotiate such complex funding systems [70]. Local organisations with limited capacity also face intense local competition in applying for the limited funding available [85]. Garside et al. (2020) further noted that third-party funders’ goals are generally aligned with NHS priorities, but not necessarily with the specific needs of local communities, causing a potential disconnect between the NHS and locally provided GSP [70].

Our findings confirm some of previous studies and mainly present the lack of funding, and competitiveness in the funding process. Frontline service providers (VCSEs) reported that they are experiencing increased demand without an associated increase in funding. They also reported difficulties such as GSP funding being awarded to GPs rather than the providers, or the need to become constituted as a legal entity to receive funding. This reflects systemic problems with current overly complex funding pathways. Lack of continuity in funding is also unhelpful for productivity and providing continuity of service for users, as well as monitoring outcomes. Interviewees also reported that services must be tailored to align with current and shifting funding priorities, and may not cover core costs, confirming that reliance on grants dictates the temporary and contingent nature of services [90]. These are ongoing challenges faced by providers:


*The main challenge is the constant fundraising that has to go on in the background … I need a funder to pay for the Big Yellow storage to buy me a new office chair [when] my chair breaks, to repair the printer to … pay the fundraising and help with marketing and just these ongoing core costs. That’s a really big challenge.*
(National, Private Company, Lead)

COVID-19 has further exacerbated these financial challenges. Many organisations found it difficult to get funding due to investment priorities shifting into other projects or sectors, in addition to the challenges of increased costs of supplies and its knock-on effects:


*The cost of supplies has gone up, so that … causes another difficulty in terms of planting through the autumn, not only the cost but potentially the supplies. So, I know that one of the tourist locations had to replace some of the gravel with woodchip … because they couldn’t get hold of certain supplies … We might have to adapt and come up with contingency plans in terms of the cost, especially when … it’s gone up 40% in value.*
(Regional Office, Federation of Charities, Project Coordinator)

### 3.4. Access to Natural Spaces

Two recent England-wide studies show persistent deficits in access to green space [91], especially in areas of deprivation [92]. Even where there is more green space in areas of higher deprivation [93], such green space is not necessarily of high quality [52,94,95], or not perceived as such [96].

Robinson et al. (2020) found that the presence of green space and abundance within close proximity (100 and 250 m) to GP surgeries is associated with the number of green social prescribing (GSP) providers [71]. They also identified further factors that limit green space access: landowner permission, transport costs, and the difficulties some people may experience in actually leaving their homes. Lack of access to green space is likely to act as a significant barrier to the setting up of new community-led programmes to engage people with nature. Land use conflicts have also been identified, including complexities in land ownership [83], conflicting public and private land use interests [97], commercial and citizens’ interests [85], and competition with other urban land uses [68]. Local authorities play a vital role in securing access to green spaces for local organisations and determining land use [69] or may impede progress by failing to deliver on promised community asset transfers [70]. Site restrictions can also constrain project design, development, and management [69,83,98].

In our study, national-level stakeholders also noted the inequalities in accessing natural space. Local organisations and communities indicated that it is difficult to get permission to use sites to start NbS projects. Issues with facilities (e.g., broken, lack of) may also impede the setting up of services. Below is an example of how a potential project was abandoned due to a land access issue:


*They [wanted] to plan things there and that was going to engage the local school kids … the complexity of trying to find out who owned and … how they could get permission … was so difficult that they just [ended] up giving up. I think for nature based projects that’s a real [difficulty], accessing lands.*
(National, Charity, Deputy CEO)

During COVID-19, some sites were closed, temporarily or permanently. At other sites, the requirement for green space maintenance increased due to the damage caused by intense usage:


*Our parks and green spaces have essentially been battered over the last 12 months because of the vast increase in usage, particularly in unseasonal … usage of these spaces. So, they will require greater levels of investment … management and maintenance.*
(National, Charity, Project Coordinator)

### 3.5. Engagement

Citizen engagement is essential to align NbS with local environmental, physical, and social contexts. However, it is often insufficiently prioritised or is considered too difficult to deal with consistently, proactively, and inclusively [68,71,78,82]. For example, participants in NbS projects may lack diversity in terms of ethnicity, age and income levels [78]. Potential reasons include the general public lack of environmental awareness, lack of funding to support engagement processes, and uncertainty about the tangible results [68]. Garside et al. (2020) also mentioned that GPs expect patients to participate actively in GSP, rather than act as passive recipients of a drug regime [70]. However, potential users’ personal circumstances may hinder such engagement, e.g., being in poor mental health or lacking the confidence to engage in the interventions, or having other day-to-day living pressures that prevent active engagement [70,71].

Similarly, stakeholder participation becomes especially pertinent when aiming to better connect people with their local natural resources [99]. However, engaging stakeholders throughout long-term projects is also considered a challenge [69,100]. Furthermore, due to dysfunctional communication between service providers and the referral sector, GPs and link workers may lack knowledge about locally available GSP services and how to access them; this is highlighted as an information availability barrier, which will prevent referrals [70].

In our study, difficulties in local engagement refer to how stakeholders understand and respond to diverse locally valued activities, and local needs (e.g., people who are digitally excluded, have disabilities, are from different cultural backgrounds and/or are experiencing high levels of deprivation). Reflecting on the general population’s involvement in NbS, there remains a need to develop effective engagement methods for reaching the people in the most need. These local engagement difficulties are recognised both locally and nationally, focusing on issues such as:


*What reaches the people in most need and who they are and how to connect with them, and what will be of interest to them.*
(National, Interest Group with Charitable Status, National Lead for the Natural Environment)

Locally available services not being known to local people, GPs or link workers, partly because the information is not kept up-to-date, was identified as another barrier by interviewees. Conversely, it is also difficult for organisations to share and update their services for wider audiences as effective systems are lacking. This information gap existed before the pandemic, and still persists:


*A directory is not going to work because if I write down today what all of my members … what their provision is, tomorrow will be different and I’ll have to update it again … Most of the ones that [are] attempted locally require the organisations to update themselves, but that fails because let’s say the organisation goes bust, there’s [no one] to update it. So … keeping a single directory of what’s available is very difficult.*
(National, Coalition of Health and Social Care Charities, Policy and Strategy Consultant)

Interviewees reported that the most obvious engagement challenge after the onset of COVID-19 was that the pandemic further exacerbated financial and health inequalities, and caused more people to experience mental health issues, making it harder for those people to participate in nature-based activities and services. Some people remain worried about going out and interacting with others, and it can be difficult to engage people remotely. Interviewees also mentioned the challenges of communicating with young people and people with specific needs. An interviewee expressed their perspective on the increased challenges of local engagement due to COVID-19:


*Particularly in the work that I do on loneliness and social isolation, the organisations that I’ve spoken to, [have seen] people whose own nest has become even more entrenched in the period of the pandemic.*
(National, Coalition of Health and Social Care Charities, Policy and Strategy Consultant)

This observation aligns with evidence in the literature that pandemic-related psychological effects make it more difficult to persuade some people to get outside [101].

### 3.6. Institutional and Organisational

Transdisciplinary projects—such as NbS and GSP, which often have to bridge multiple disciplines and sectors—need clear leadership to build trust and consistent relationships, and to transcend institutional boundaries, a lack of which will result in a lack of clarity concerning roles and responsibilities [76]. In addition, Sarabi et al. (2021) identified a lack of leadership in relation to NbS projects from municipalities and a lack of motivation from the private sector to jointly take responsibility [78]. Another organisational challenge is internal governance: an inflexible hierarchical organisational structure will cause difficulties in engaging key municipal departments and hinder the exercise of managerial discretion. These will lead to the implementation process becoming slow and overly bureaucratic [69,78], as well as difficulty in collaboration and fragmentation within the municipal organisation [102]. The overwhelming demand for GSP is challenging for service providers, as many of them are small organisations and are increasingly expected to support clients with complex needs [70]. That demand revealed capacity and human resource issues, including insufficient volunteer capacity [70,80], and high staff turnover [69,78].

Interviewees from a local authority and a social prescribing service also confirmed that demand for social prescribing has moved beyond the system’s capacity, and referees may be required to wait for extended periods and consequently become disengaged. Interviewees also reported issues related to capacity and human resources. Capacity issues include people being referred to organisations with specific needs beyond the scope of the services provided, and communities lacking the capacity to produce plans and to get involved with developing projects themselves. It also refers to GPs having very limited capacity for dealing with each patient, which may lead to inappropriate referrals and lack of support for connecting patients with GSP providers, making it difficult for some people to participate in activities on offer. These challenges reveal the need for increased financial and human resources, as well as greater capacity development support, for local organisations to scale up and deliver appropriate services, and GPs to make appropriate referrals. In the words of one of our interviewees:


*The difficulty for us is taking on more and more referrals, and without proper services really putting in either resource or finance into the service that they’re trying to prescribe to … there is a capacity issue in the size of [the] social prescribing model, from our point of view.*
(Local, NbS Provider, Founder and Director)

Human resource issues included ageing staff approaching retirement and a heavy reliance on retirees as volunteers, a lack of local engagement officers (from a national level organisation), as well a lack of regular volunteers, and difficulties in recruiting new volunteers. Here, an interviewee explains how local organisations are struggling with human resources issues:


*We’re trying to develop our volunteer capacity to do more in the local area with less. So, I think resources and capacity are probably the two issues that I think will be a challenge [to improve or increase the use of nature for social prescribing or for public health and wellbeing purposes].*
(Local, Local Authority, Community Development Officer)

Fixsen and Barrett (2022) found that volunteering declined due to the upheavals and health concerns caused by the pandemic [101]. Our study confirmed that after the onset of COVID-19, organisations lost skilled staff, and found it difficult to re-engage volunteers due to some being elderly, or not wishing to travel. Both staff and volunteer issues can result in projects becoming unsustainable and stopping altogether. This situation has not necessarily resolved itself now that lockdown periods appear to be over, something that was echoed by interviewees at all levels, who said that local organisations continued to experience heightened human resources and capacity issues, together with overwhelmingly increased demand.

COVID-19 restrictions posed considerable challenges to providers. Some projects stopped due to COVID-related issues, and it has taken a long time for them to restart. Services that remained in operation experienced potential users having difficulties in travelling, restricted face-to-face interaction, a necessary reduction in activities, and the need to operate in a COVID-19 secure way, which led to greater preparation times, all of which significantly reduced the opportunities to engage people effectively.

### 3.7. Coordination

NbS requires interdisciplinary coordination and knowledge for planning, designing, implementing, maintaining and monitoring projects [83,103]. However, as Dorst et al. (2022) mentioned, joint action for development and investment in NbS is lacking [68]. To implement NbS, coordination between different organisations is widely considered important but challenging [104,105,106].

Community groups wishing to set up new NbS to connect people with nature in their local area, whether they are targeting the general population or specific groups, or wanting to offer GSP, have to negotiate complex systems involving potential sites, funders, stakeholders, and legal and regulatory frameworks, as well local politics, custom and practice [80]. The complexity of the systems that NbS have to engage with also raises a related coordination issue: sectoral silos are often a fundamental barrier for different stakeholders to effectively communicate and coordinate [77,78,107]. Redondo Bermúdez et al. (2022) pointed out that different sectors and stakeholders operate according to their own agendas, timeframes and values, and the clarity of ownership, responsibility, and relationships among them are not clear [69]. Negotiating this complexity is very challenging in the absence of a coordinated approach amongst all the organisations and sectors involved.

This study also identified coordination challenges related to the difficulties of different sectors partnering or collaborating to implement NbS. Interviewees discussed the difficulties organisations face in establishing partnerships with the NHS to provide green social prescribing (GSP). For example, interviewees from both VCSE and healthcare perspectives all highlighted the lack of coordination between GSP and NHS systems. This is in line with previous findings that green health is not fully embedded into core development strategies by health, social care, local authorities, and the third sector [80]. Therefore, together with the pressure faced by the health system, COVID-19 also makes the coordination between organisations and health and social care systems more difficult. Here, a project lead discusses the challenges in partnering with the NHS:


*We partner with… either your local GP practice or … we are working with hospital departments to include them in the project in different ways. But … it’s not a partnership between us and the NHS, but it’s a service that is designed around the NHS to improve public health, but … it’s incredibly difficult … you wouldn’t think that [with] it being so easy for the NHS to take part in it … it would be this hard.*
(National, Charity, Green Space Communities Lead)

Previous research identified that small NbS providers find it difficult to gain a foothold in local social prescribing systems, as they may be unaware of how to engage with those systems and devote time to making connections with local link workers [70,108]. In this study, coordination challenges include a lack of, or difficulties in, communication between local people, local authorities, VCSEs, and local health and social care systems, as well as difficulties in maintaining networks after project completion. This lack of communication can lead to difficulties in empowering local communities—including those with high levels of deprivation—to get involved, and some interviewees felt unsure about how local authorities could support local communities to instigate projects. A participant from a local NbS provider described these barriers in communication:


*My role has almost been to communicate … how can it benefit their organisation [a newer and smaller GSP provider], how they fit into the whole picture. … That’s been part of my role as a trustee for [Local Umbrella Organisation], but it’s been really frustrating that at a strategic level, I guess, everyone understands that the communication hasn’t filtered down.*
(Local, NbS Provider, Founder and Director)

### 3.8. GSP Referral and Service Issues

In addition to the coordination issues already discussed, referral routes and criteria are key challenges in implementing social prescribing [109]. Garside et al. (2020) reported a series of issues in the complex GSP referral process: the plurality of different referrers causing confusion, opaque systems, unclear language and entry points, and geographic and sectoral variation in link workers’ ability to approach patients and to access patient records [70]. Conversely, while GPs value feedback on outcomes, this is considered a further administrative burden by link workers. Fullam et al. (2021) also noticed that NbS providers need to understand their local referral pathways, and the roles of various sectors, which may vary locally [108]. Lack of awareness of NbS benefits and lack of service quality assurance may result in difficulties in engaging GPs, as they may be unable or unwilling to give green social prescriptions [71,80].

Participants in this study reported the referral process is overly bureaucratic and inconsistent which limits the wider take-up of GSP. An interviewee from a GSP provider organisation found it difficult to provide appropriate services when patients were referred to their organisation without relevant patient information, as they needed to spend time establishing the social determinants of that patient’s health. The complexity of the referral process is noted as inefficient:


*That bureaucratic sense of having to go through a referral process then wait for an assessment … a lot of that can be resolved cost effectively by better knowledge.*
(Local, Authority, Lead Commissioner)

This study confirmed a longstanding challenge around quality assurance in GSP service provision [110]. Social prescribing services—including those offering nature-based activities—have difficulty in monitoring quality and outcomes and may therefore not be able to demonstrate efficacy. GPs’ uncertainty about service quality is not only a barrier to making more GSP referrals, but also prevents wider promotion and acceptance of GSP services.


*The outcomes are not monitored … so you can’t evaluate the effectiveness or the impact of social prescribing because you’re not measuring the same thing. So, there needs to be an evidence base attached to social prescribing that’s accepted by practitioners.*
(Local, Not-for-profit Housing Association, Social Inclusion Manager)

## 4. Conclusions

This paper has further explored challenges to the implementation of NbS and GSP developing on pre-existing knowledge, focusing on the barriers faced by community-led NbS, especially after the onset of the COVID-19 pandemic. We present a revised typology of barriers, incorporating previous classifications and the emergent barriers identified in our stakeholder interviews; these barriers are the reality faced by current stakeholders, the solutions to which are not yet reflected in policy or practice. Our typology can be used in future research and practice by multiple disciplines and sectors to understand their roles in implementing community-led NbS, and to identify potential ways to work more collaboratively in this process.

There is wide consensus with regard to the lack of awareness of the efficacy and cost of NbS amongst the general public and key stakeholders, including service users and healthcare professionals, borne out by the findings from this study. In line with previous research, our study also highlights the issues of reliance on and competition for sporadic and unsustainable funding, leading to reliance on volunteers’ continuity and capacity, and consequential difficulties in providing long-term services.

As noted in previous research, inequality in access to green space, and maintaining green space quality, are nationwide challenges. In this context, this study highlighted that local authorities play a key role in permitting local organisations access to green space and must make the processes involved straightforward and transparent. As well as accessing land for NbS there are problems knowing what form those nature-based projects should take once access is granted. With a specific focus on community-led NbS projects, this study further highlighted the lack of effective engagement methods to understand and respond to the diversity of local needs.

Multi-sectoral coordination is essential for implementing community-led NbS projects. Previous research has explored the complexity of the systems involved; this study highlights the lack of communication between sectors, leaving actual and potential NbS providers unsure as to the processes and resources available, and their potential partners and supporters unclear about what assistance is needed locally. Furthermore, locally available services are not always known to users, GPs or link workers. The development of a platform is needed to integrate resources and an efficient referral system that is workable and sustainable for health and social care referrers and service providers, as well as ensure that patients are referred to suitable services. Our study found that GSP providers are currently placed in the awkward situation of relying on the NHS to make referrals while being poorly connected with NHS systems themselves. The difficulties in collaborating with the NHS are echoed by issues in the referral process, which is overly bureaucratic and complex. Consistent with previous findings that monitoring service quality and outcomes, and balancing demand and capacity, are also longstanding challenges for GSP promotion, this paper further noted the difficulties in GSP adjusting to pandemic-related restrictions.

This study has provided insights into the additional challenges caused by the COVID-19 pandemic. The pandemic saw increased wear and tear on urban green spaces, and this study has highlighted the difficulties in maintaining the condition of green space due to this intensity of usage, and in reopening sites and restarting NbS projects following the onset of COVID-19. This paper also reveals that financial and health inequalities were exacerbated, NbS providers were facing an overload in demand, and more difficulties in engaging people under COVID-19 restrictions, alongside the loss of human resources caused by COVID-19. More work is needed to develop and test a range of methods for engaging a diversity of local population groups with the highest levels of need.

To conclude, more research to provide convincing evidence, to make policy-makers aware of the importance and urgency of NbS, to secure political will and long-term commitment for systemic change, and prioritise funding support, all of which would support the development and mainstreaming of NbS in policy and practice. If existing policy commitments related to expanding green infrastructure, setting new quality standards, and moving the NHS model from treatment to prevention, are to be met, and GSP potential fulfilled, then the VCSE sector’s potential to make a vital contribution must be recognised, and systemic challenges addressed. The nature’s way project found that the VCSE sector holds a wealth of knowledge and experience in addressing the needs of local communities, but that systemic challenges currently prevent this potential from being realised.

## Figures and Tables

**Table 1 ijerph-19-14906-t001:** Participants’ roles and organisations.

Organisation Level	Type of Organisation	Job Description
National	Private Company	Nature Lead
National	Funding Body	Head of relevant section
National	Interest Group with charitable status	National Lead for relevant department
National	Built Environment Professional Body	President
National	Coalition of Health and Social Care Charities	Policy and Strategy Consultant
National	Charity	Green Space Communities Lead
National	National Park Authority	Interpretation and Outreach Manager
National	Environmental Charity	Head of Health and Education
National	Private Company	CEO and Founder
National	Charity	Project Coordinator
National	Non-departmental Public Body	Senior Adviser
National	Charity	Manager
National	Charity	Deputy CEO
National (Regional office)	Federation of Charities	Project Coordinator
Regional	NbS Provider	Chief Executive
Local	University	Wellbeing Coordinator
Local	Not-for-profit Housing Association	Customer Voice
Local	Local Authority	Lead Commissioner
Local	NbS Provider	Founder and Director
Local	NbS Provider	Chairperson
Local	Local Authority	Community Development Officer
Local	NHS Primary Care	Social Prescriber
Local	Not-for-profit Housing Association	Social Inclusion Manager
Local	Not-for-profit Housing Association	Social Prescribing Manager
Local	Local Authority	Team Leader
Local	Local Authority	Project Officer

**Table 2 ijerph-19-14906-t002:** Existing and emergent challenges to implementing NbS.

Categories	Challenges	COVID-19 Caused Challenges
Perception and knowledge	**Lack of understanding and awareness of NbS and their benefits**	
	**Lack of experience and knowledge**	
	**Lack of consistent terminology**	
	**Path dependency**	
	**Lack of awareness of the cost of NbS**	
Political	Lack of sense of urgency	
	Lack of political will and long-term commitment	
	Lack of supportive legal and policy frameworks	
	**Lack of national level leadership**	
	**Moving the medical model from cure to prevention**	
Financial	Complex funding system	**Investment priorities shifting**
	Mismatch between funding priorities and local needs	**Increased cost of supplies and knock-on effects**
	**Lack of sustained or dedicated financial resources**	
	**Competitiveness in the funding process**	
Access to natural spaces	Land use conflicts	**Site closed temporarily or permanently**
	Site restrictions	**Green space damaged by intense usage**
	**Deficits in access to green space**	
	**Get permission to green space**	
	**Facility issues**	
Engagement	Insufficient citizen engagement	**Exacerbation financial and health inequalities**
	Initiating and sustaining stakeholder engagement	**Difficulty in engaging people remotely**
	**Lack of knowledge about current locally available services and how to access them amongst stakeholders and the public**	
	**Stakeholder’s lack of knowledge about local needs and contexts**	
	**Lack of effective engagement methods**	
Institutional and organisational	Lack of leadership	**Exacerbation of human resource and capacity issues**
	Internal governance	**Facing aggressively increased demand**
	**Facing overwhelming demand**	**Coping with COVID-19 restrictions**
	**Capacity and human resource issues**	
Coordination	The complexity of stakeholder systems	**Difficulty in the shift to remote working**
	**Difficulty in establishing partnership with the NHS**	
	**Lack of communication across all sectors**	
	**Difficulty in empowering local communities**	
GSP referral and service issues	**The complexity of the GSP referral process**	
	**Lack of quality assurance**	

Consistent challenges identified in both the literature and this study are in bold and highlighted in grey, challenges emergent from this study are in bold.

## Data Availability

Due to the ethically and politically sensitive nature of the research, participants of this study did not agree for their data to be shared publicly. Please contact the project PI at qian.sun@rca.ac.uk to discuss the underlying data.

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
