# Peer review of "Negotiating Complexity: Challenges to Implementing Community-Led Nature-Based Solutions in England Pre- and Post-COVID-19"

_ijerph, 2022, doi:10.3390/ijerph192214906_

Round 1

Reviewer 1 Report

This paper is very valuable, and I can immediately see myself returning to it often, and recommending it to my postgraduate students working in the area of well-being.  There has been a lot of general discussion about green space, well-being and COVID-19, and this paper provides an accessible and comprehensive insight into the issues.  This is augmented with specific examples that illustrate the challenges of the intersections of the spheres of well-being, green space provision, green space utilisation, and health aspects such as COVID-19.

 The results and discussion are very clear, and the identification of themes provides a vivid set of issues to emerge from the interviews.  There is the potential for the study to be replicated elsewhere, especially given the clear description of methods.  It already has me wondering how much the cultural and political contexts of different countries might influence the types of challenges that are perceived – not to mention wider aspects related to things such as urban densities etc in relation to green space provision.  The paper is therefore of value in the way in which it sows many seeds for further research.

A fewthings to enhance readability and clarity:

One is the extensive use of abbreviations for many of the terms.  Some of these are specific to the UK context, and for readers not familiar with them it can be a bit frustrating going back to check what a particular abbreviation meant.  And some are from the language of health – CMD GSP etc, which is not surprising given the journal this will be published in.  But it is likely to be a paper that is read by many landscape architects (like me!) who are not so familiar with the terminology.  If possible it would be useful to either occasionally use the full term throughout, to remind readers what it means, or to include a short glossary of abbreviations. 

The second point to enhance clarity would be to expand a little on the section on p.3, lines 103-109.  It is explained that deprived areas have less green space.  And then the statement is made that “US communities with the least green space nearby were the ones most impacted by COVID-19.”  This almost reads as a causal relationship … and what is missing is an explanation before this statement that not only is there inequity in the provision of greenspace, but there was also inequity in the way in which COVID-19 had a more devastating impact on lower socio-economic groups.  Which is to say there is an intersectional situation that includes both access to health care (and all that goes with that in terms of nutrition, doctors visits, vaccination etc) AND access to good quality green space. 

I suggest the paragraph that encompasses lines 132 -137 could be restructured for clarity. At present there is a huge statement within the dashes that explains the project.  This in itself is clear, but given the entire paragraph is one sentence it does become a bit convoluted. 

It isn’t clear what is meant by [Project name] in lines 166, 192 and 589 – perhaps I missed an explanation of what this relates to, but I can’t work out what is intended.  If it is a way of referring to the data in a specific way depending on a project name, it would be useful to include an example of what this looks like.  Maybe it is to do with ethics and anonymity, as it is noted that some potentially identifying labels have been replaced by generic terms in square brackets elsewhere in the document?

It is good to see the inclusion of citations from the interviews, but these are a little confusing in terms of the formatting.  Either using quote marks or indenting these quotations would make it clearer in terms of the material which comes from the interviews verbatim. In some cases the cited text is in italics which makes it easier to identify, but at present it is inconsistent. 

Author Response

Dear reviewer,

We appreciate the time and effort that you have dedicated to providing your valuable feedback on my manuscript. Below are our responses regarding your comments.

Comment 1:
One is the extensive use of abbreviations for many of the terms.  Some of these are specific to the UK context, and for readers not familiar with them it can be a bit frustrating going back to check what a particular abbreviation meant.  And some are from the language of health – CMD GSP etc, which is not surprising given the journal this will be published in.  But it is likely to be a paper that is read by many landscape architects (like me!) who are not so familiar with the terminology.  If possible it would be useful to either occasionally use the full term throughout, to remind readers what it means, or to include a short glossary of abbreviations.

Response:
Thank you for pointing this out. We are aware that the extensive use of abbreviations might be frustrating for readers. Therefore, the abbreviation of CMD has been replaced by the full term (common mental disorders). The abbreviation of GSP has been occasionally replaced by the full term throughout the manuscript.

Comment 2:
The second point to enhance clarity would be to expand a little on the section on p.3, lines 103-109.  It is explained that deprived areas have less green space.  And then the statement is made that “US communities with the least green space nearby were the ones most impacted by COVID-19.”  This almost reads as a causal relationship … and what is missing is an explanation before this statement that not only is there inequity in the provision of greenspace, but there was also inequity in the way in which COVID-19 had a more devastating impact on lower socio-economic groups.  Which is to say there is an intersectional situation that includes both access to health care (and all that goes with that in terms of nutrition, doctors visits, vaccination etc) AND access to good quality green space.  

Response:
Thank you for this suggestion. We have added content at page 3 about health inequality and deprivation to clarify this point.

Comment 3:
I suggest the paragraph that encompasses lines 132 -137 could be restructured for clarity. At present there is a huge statement within the dashes that explains the project. This in itself is clear, but given the entire paragraph is one sentence it does become a bit convoluted.  

Response:
The paragraph has been divided into several sentences for clarity. Considering the other reviewer’s suggestions, some new contents have been added into this paragraph to help the reader understand the project's relevance to the current study.

Comment 4:
It isn’t clear what is meant by [Project name] in lines 166, 192 and 589 – perhaps I missed an explanation of what this relates to, but I can’t work out what is intended.  If it is a way of referring to the data in a specific way depending on a project name, it would be useful to include an example of what this looks like.  Maybe it is to do with ethics and anonymity, as it is noted that some potentially identifying labels have been replaced by generic terms in square brackets elsewhere in the document?

Response:
The [project name] refer to the Nature’s Way project which this paper is based on. We are sorry that they were missed from the check before submitting. We have changed them from [project name] to Nature’s Way project.

Comment 5:
It is good to see the inclusion of citations from the interviews, but these are a little confusing in terms of the formatting.  Either using quote marks or indenting these quotations would make it clearer in terms of the material which comes from the interviews verbatim. In some cases the cited text is in italics which makes it easier to identify, but at present it is inconsistent.  

Response:
We are sorry about the formatting issues. All the citations from the interviews are now indented and formatted in italics. 

In addition to the above comments, we also corrected some minor spell mistakes.

We look forward to hearing from you regarding our revised manuscript and to respond to any further questions and comments you may have.

Best regards,
All authors

Reviewer 2 Report

Dear Authors

Thank you for the opportunity to review this manuscript on an important area of research. From reading the abstract, you report what appears to suggest a qualitative study of 26 stakeholder interviews to inform the challenges facing community-led nature-based solutions in the UK within the context of the COVID-19 pandemic. Eight categories of challenges were identified. The abstract does not reflect the manuscript which introduces a literature review without explanation of study design and methods, and a qualitative study, again with insufficient detail of study design. As such, much revision is required.

Introduction: At the end of the last paragraph, you refer to using the findings of the Nature’s Way Project, please provide background information to help the reader understand the projects relevance to the current study.

Methodology: Why have the authors introduced a literature review without explanation of methods? From the abstract, I interpret your study to be qualitative. Please re-write including methods.

For the qualitative study, please explain how participants were approached. For example, were they emailed etc., and did you have any participants refuse to take part or drop out?

For the interviews, please explain the setting for the data collection. Where did the face-to-face interviews take place? Also, explain the setting for the telephone interviews. 

Within the data Analysis, there is insufficient evidence of the development of the coding categories, sub-themes, and themes. Please include a coding tree or equivalent as a supplementary file.

In addition, the results lack participant quotations to illustrate the themes. Can the authors provide a table or supplementary evidence of participant quotations to support each theme? Also, why do the authors emphasize the pre and post COVID-19 period within the title and abstract when some themes do not clearly provide sufficient evidence of this. Further, why do the authors omit the discussion? I would suggest, re-writing the intro to specifically frame the problem with shorter paragraphs and then include a discussion section to highlight the key findings in the context of the literature.

Author Response

Dear reviewer,

We appreciate the time and effort that you have dedicated to providing your valuable and insightful feedback on our manuscript. Below are our responses regarding your comments.

Comment 1:

Introduction: At the end of the last paragraph, you refer to using the findings of the Nature’s Way Project, please provide background information to help the reader understand the projects relevance to the current study. 

Response:

Thank you for pointing this out. We have added new content to introduce the project and its relevance to the current study in the section near the end of the introduction.

Comment 2:

Methodology: Why have the authors introduced a literature review without explanation of methods? From the abstract, I interpret your study to be qualitative. Please re-write including methods. 

Response:

Thanks for your insightful comments regarding the methodology section. We apologise if our expression about the literature review has caused a misunderstanding. We have added content in section 2.2 to further explain the traditional literature review that we did to establish the context for the study, and made corresponding changes in the abstract.

Comment 3:

For the qualitative study, please explain how participants were approached. For example, were they emailed etc., and did you have any participants refuse to take part or drop out?

Response:

Thanks for your comment about how we approached the participants. We have added relevant content in section 2.2 to explain that stakeholders were approached by emails in the first instance. There were 8 stakeholders who refused to participate but referred us to an alternative contact within their organisations, and no participants dropped out.

Comment 4:

For the interviews, please explain the setting for the data collection. Where did the face-to-face interviews take place? Also, explain the setting for the telephone interviews. 

Response: 

We have added content to explain the setting for data collection as follows: ‘The interviews were conducted by experienced research associates from July 2021 to February 2022, remotely via Zoom at a quiet office or home setting due to the pandemic and the fact that many stakeholders were dispersed around England, or face-to-face on site at the participants’ NbS projects when possible.’ 

Comment 5:

Within the data Analysis, there is insufficient evidence of the development of the coding categories, :sub-themes, and themes. Please include a coding tree or equivalent as a supplementary file. In addition, the results lack participant quotations to illustrate the themes. Can the authors provide a table or supplementary evidence of participant quotations to support each theme? 

Response:

Thank you for this point. In considering the length of the paper, we did not add quotations for each of the themes. We will submit a supplementary document that includes the coding themes, sub-themes, and corresponding quotation examples for each main theme.

Comment 6:

Also, why do the authors emphasize the pre and post COVID-19 period within the title and abstract when some themes do not clearly provide sufficient evidence of this. 

Response:

Thank you for your thoughtful comments on this. The reason that some themes did not provide sufficient evidence of the challenges post COVID-19 is that our data only reflect changes in certain categories, they are within the categories of Financial, Access to natural spaces, Engagement, Institutional and organisational.

Comment 7:

Further, why do the authors omit the discussion? I would suggest, re-writing the intro to specifically frame the problem with shorter paragraphs and then include a discussion section to highlight the key findings in the context of the literature.

Response:

Thank you for this suggestion. Our discussion is combined with the results. We did have an earlier version with independent results and discussion sections, but that will significantly increase the length of the manuscript. As often in the case of qualitative research, the results are discussed as they are detailed. In order to reduce the repetition and the length of the manuscript, we combined our results and discussion.

In addition to the above comments, we corrected some minor spelling mistakes, and added some content that was suggested by other reviewers.

We look forward to hearing from you regarding our revised manuscript and to responding to any further questions and comments you may have.

Best regards,

All authors

Reviewer 3 Report

Dear Authors,

I found your work very interesting. Although the topic is not innovative, as there are few studies concerning the beneficial role of physical activity in a natural environment, targeting a diversified group of study stakeholders representing environment, health, and social care sectors at national and local level, and focusing on institutionalized Green Social Prescribing system, fills the research gap. The structure of the article is clear. Methods are properly applied. The results and discussion are well written.

Perhaps it would be valuable to elaborate a little bit more in the Introduction on the background of Green Social Prescribing, referring to its origins (Social Prescribing), explaining in which other countries it could be encountered? Does it refer to an individual or to a household? Does it differ in the United Kingdom (how does it work in four different health systems England, Wales, Scotland, Northern Ireland?).

I would recommend broadening teh reference list with:

•             Younan, H. C., Junghans, C., Harris, M., Majeed, A., and Gnani, S. (2020). Maximising the impact of social prescribing on population health in the era of COVID-19. J. R. Soc. Med.113, 377–382. doi: 10.1177/0141076820947057

•             The Health Foundation (2014). The Four Health Systems of the United Kingdom: How do they Compare?. London: The Health Foundation.

•   Global Social Prescribing Alliance (2022). Global Social Prescribing Alliance: Supporting SDG3 Good Health and Wellbeing. Available online at: https://www.gspalliance.com

•             NHS England (2022a). Delivering Universal Personalised Care. Available online at: https://www.england.nhs.uk/personalisedcare/social-prescribing/

•             NHS England (2022b). Green Social Prescribing. Available online at: https://www.england.nhs.uk/personalisedcare/social-prescribing/green-social-prescribing/

I believe that your work will be a valuable contribution for the Journal and for promotion of the benefits provided by physical activity outdoors.

Best regards,

The reviewer

Author Response

Dear Reviewer,

We are grateful to your insightful comments on our manuscript. Below are our responses regarding your comments and suggestions.

Comment 1:

Perhaps it would be valuable to elaborate a little bit more in the Introduction on the background of Green Social Prescribing, referring to its origins (Social Prescribing), explaining in which other countries it could be encountered? Does it refer to an individual or to a household? 

Response:

Thank you for your suggestions and suggested references. We have added some content about social prescribing with relevant references, and the countries it has been implemented at the start of the introduction.

Comment 2:

Does it differ in the United Kingdom (how does it work in four different health systems England, Wales, Scotland, Northern Ireland?).

Response:

Thanks for pointing this out. We are aware that social prescribing might be different in different health systems. It would have been interesting to explore this aspect. But it can be very complex to compare and could add the length of this long manuscript. We wondered if we could change the title from ‘UK’ to ‘England’ to make it more clear that the data we collected reflected the situation in England.

We look forward to hearing from you regarding our revised manuscript and to respond to any further questions and comments you may have.

Best regards,

All authors

Round 2

Reviewer 2 Report

Thank you for the opportunity to read the updated manuscript.  The updated abstract now better informs the reader of the included literature review, as does the data collection section (2.2) of the manuscript. The authors have addressed all comments and I am happy to accept the updated manuscript in its present form.